# Carvacrol Microemulsion vs. Nanoemulsion as Novel Pork Minced Meat Active Coatings

**DOI:** 10.3390/nano13243161

**Published:** 2023-12-18

**Authors:** Konstantinos Zaharioudakis, Eleni Kollia, Areti Leontiou, Dimitrios Moschovas, Andreas Karydis-Messinis, Apostolos Avgeropoulos, Nikolaos E. Zafeiropoulos, Efthymia Ragkava, George Kehayias, Charalampos Proestos, Constantinos E. Salmas, Aris E. Giannakas

**Affiliations:** 1Department of Food Science and Technology, University of Patras, 30100 Agrinio, Greece; zacharioudakis.k@upatras.gr (K.Z.); aleontiu@upatras.gr (A.L.); effierag@yahoo.com (E.R.); gkechagi@upatras.gr (G.K.); 2Laboratory of Food Chemistry, Department of Chemistry, National and Kapodistrian University of Athens Zografou, 15771 Athens, Greece; elenikollia@chem.uoa.gr (E.K.); harpro@chem.uoa.gr (C.P.); 3Department of Material Science and Engineering, University of Ioannina, 45110 Ioannina, Greece; dmoschov@uoi.gr (D.M.); a.karyd@uoi.gr (A.K.-M.); aavger@uoi.gr (A.A.); nzafirop@uoi.gr (N.E.Z.)

**Keywords:** carvacrol, microemulsion, nanoemulsion, active coatings, nanostructure, pork meat preservation, antioxidant activity, antibacterial activity

## Abstract

Carvacrol is well documented for its antibacterial and antioxidant effects. However, its high volatility has directed researchers toward nanoencapsulation technology according to bioeconomy and sustainability trends. This study examined and compared free carvacrol (FC), carvacrol microemulsion (MC), carvacrol microemulsion busted with chitosan (MMC), and carvacrol nanoemulsions (NC) as active coatings on extending minced pork meat shelf life at 4 ± 1 °C for 9 days, focusing on microbiological, physiochemical, and sensory characteristics. The research involved pre-characterizing droplet sizes, evaluating antioxidants, and determining antibacterial efficacy. The results demonstrated that NC with a 21 nm droplet size exhibited the highest antioxidant and antibacterial activity. All coatings succeeded in extending the preservation of fresh minced pork meat in comparison to the free carvacrol sample (FC). The NC coating showed the highest extension of minced pork meat preservation and maintained meat freshness for 9 days, with a lower TBARs of 0.736 mg MDA/Kg, and effectively reduced mesophilic, lactic acid, and psychotrophic bacterial counts more significantly by 1.2, 2, and 1.3 log, respectively, as compared to FC. Sensory assessments confirmed the acceptability of NC and MCC coatings. Overall, the carvacrol-based nanoemulsion can be considered a novel antioxidant and antimicrobial active coating due to its demonstrated higher efficacy in all the examined tests performed.

## 1. Introduction

Nowadays, the need to keep foods fresh and without spoilage for as many days as we can has directed food technology toward the investigation of novel antimicrobial/antioxidant systems to preserve foods. Simultaneously, sustainability and circular economy trends drive the replacement of food chemical additives with natural abundant antimicrobial and antioxidant agents. Antimicrobial agents are necessary for food preservation to prevent the growth of harmful microorganisms that can cause spoilage, reduce the shelf life of food products, and pose health risks to consumers [1,2]. Natural compounds like carvacrol, found in essential oils of various herbs, have been shown to have effective antimicrobial activity in food systems, including meat products [3,4]. Carvacrol is a natural phenolic monoterpene that is synthesized from the mevalonic acid pathway and is commonly found in herbs and spices such as oregano, thyme, and marjoram [5]. It has been shown to possess potent antimicrobial properties against various microorganisms, including bacteria and fungi, making it a promising natural alternative to synthetic preservatives in the food industry [6,7]. Carvacrol’s mechanism of action involves disrupting the cell membrane of microorganisms, inhibiting adenosine triphosphate (ATP) synthesis, and leading to cell death [8,9]. Additionally, carvacrol has anti-inflammatory, analgesic, and antioxidant effects, which make it a potential candidate for use in various applications in the pharmaceutical and cosmetic industries [10].

Recent studies have demonstrated the high antimicrobial activity of carvacrol compared to sulfanilamide against various bacteria [11]. Investigations into the potential of carvacrol for food preservation have included the addition of carvacrol in an osmotic pretreatment to extend the shelf life of a food product by 3 days [12], the development of carvacrol-loaded zein nanoparticles for active packaging in the meat industry [13], and a chitosan (CS) coating with essential oils, including carvacrol, to preserve fresh blueberries during postharvest storage [14]. These findings suggest that carvacrol is a promising natural alternative for food preservation in various forms.

Other studies have shown that controlled-release carvacrol powder can be effective for preserving small fruit, as it has been found to reduce microbial growth and weight loss, and improve sensory quality [15]. Additionally, microencapsulated thymol and carvacrol in polymer films have been found to be natural antimicrobial solutions for food packaging, which can inhibit the growth of various microorganisms. These findings suggest that carvacrol and thymol exhibit potential as natural alternative additives for food preservation in various forms [16].

However, the direct application of carvacrol to food preservation is problematic due to the hydrophobic nature of this essential oil, which affects its solubility. Additionally, for food packaging and preservation endeavors, it is crucial to consider the high volatility of carvacrol, which can be enhanced by exposure to light and heat. Such instability can diminish its effectiveness and restrict its utility in various applications [17]. To address these obstacles, encapsulation and more specific micro/nanoencapsulation techniques have been suggested as a treatment. The application of such techniques serves to enhance the bioavailability and extend the half-life of phytochemicals such as carvacrol by safeguarding the active constituents against potential degradation caused by environmental factors [18]. The utilization of encapsulation techniques minimizes the inherent instability limitations of carvacrol, thereby enhancing its efficacy because stability is of the utmost importance. Moreover, there is a growing interest in food preservation for natural antimicrobial exploitation, for example, of carvacrol, as a substitute for chemical preservatives. The utilization of encapsulation techniques, specifically micro-/nanoencapsulation, has been proposed to address various obstacles associated with solubility, stability, and alterations in sensory properties [19].

Nanotechnology, particularly in the form of nanoemulsions (NEs), offers innovative solutions to challenges in food quality and quantity by employing materials smaller than 100 nm. Designed to deliver bioactive compounds like carvacrol, NEs are composed of oil, water, and emulsifiers that form minuscule droplets, encapsulating the bioactive elements in stabilizers for enhanced structural integrity. Due to their small droplet sizes [19], NEs not only allow for improved penetration and distribution of active ingredients like carvacrol in food systems, but also enhance their solubility, stability, and bioavailability [18]. Produced through high-energy methods such as high-pressure homogenization, these nanoemulsions have proven effective in inhibiting microbial growth in various food systems, including meat products [20,21]. Their nanoscale dimensions enable the efficient delivery of active substances in reduced dosages, making them versatile for applications across the pharmaceutical, medical, and food sectors.

Recent research results have highlighted carvacrol and thymol’s effective use in food preservation and medical treatments. Powder loaded with carvacrol, which was released under a controlled rate, has proven effective in preserving small fruit, reducing microbial growth, minimizing weight loss, and enhancing sensory quality [15]. This result enhances the potential to use this essential oil as a natural, alternative, food preservation additive. Similarly, microencapsulated thymol and carvacrol in polymer films have emerged as natural antimicrobial materials for food packaging, effectively inhibiting the growth of various microorganisms. Based on such food preservation achievements, the current study has shown the advantages originating from the incorporation of low molecular weight chitosan with a microemulsion system. This result confirms reports in the literature mentioning that the chitosan–microemulsion matrix notably improves mechanical properties such as adhesiveness [20,21]. The results of various studies including the current one demonstrate the significant potential of carvacrol, thymol, and chitosan in enhancing food preservation and medical treatment efficacy [16].

Meat deterioration is a complex process influenced by factors such as lipid oxidation and microbial spoilage. Lipid oxidation is a chemical reaction involving polyunsaturated fatty acids and reactive oxygen species that leads to the formation of volatile and toxic compounds like aldehydes and ketones. These compounds not only degrade the sensory and nutritional quality of the meat but also pose health risks to consumers. Microbial spoilage is exacerbated by factors like pre-slaughter and post-slaughter handling, the bacterial flora of the animal, and processing methods. Stress before slaughter depletes muscle glycogen, leading to the production of lactic acid and changes in meat pH. A higher pH level of 6.4–6.8 results in dark, firm, and dry (DFD) meat, which is more susceptible to bacterial growth [22]. Therefore, controlling oxidation and microbial contamination is key to preserving the quality and safety of meat products.

This study aimed to conduct a comparative analysis of the preservation capabilities of free essential oils, microemulsions, and carvacrol-enriched nanoemulsions. An innovative carvacrol microemulsion fortified with chitosan (CS) was developed during this research project and its application to fresh pork minced meat preservation was investigated. The specific objective of this work was the evaluation and the identification of statistically significant differences in the meat’s shelf life and quality attributes such as microbial growth, oxidative stability, and sensory properties, following different methods of preservation over a controlled storage period of 9 days. This research intended to elucidate the relative effectiveness of each preservative approach, thereby determining the optimal formulation, based on the enhanced stability of nanoemulsions and the antimicrobial efficacy of CS and carvacrol for meat preservation.

## 2. Materials and Methods

### 2.1. Materials

Carvacrol (CAS No. 499-75-2), ethanol (CAS No. 64-17-5), TWEEN 80 (CAS No. 9005-65-6), sodium caseinate (CAS No. 9005-46-3), L-α-lecithin, soybean (CAS No. 8002-43-5), and glacial acetic acid (CAS No. 64-19-7) were purchased from Sigma-Aldrich Co. (3050 Spruce Street, St. Louis, MO 63103, USA; Tel.: 314-771-5765). Chitosan with a molecular weight of 100,000–300,000 was procured from Acros Organics (Zeel West Zone 2, Janssen Pharmaceuticalaan 3a, B2440 Geel, Belgium).

Müeller–Hinton agar plates, Müeller–Hinton broth, Tryptic Soy Agar (TSA), and sterile swabs and forceps were also acquired from Sigma-Aldrich Co. (3050 Spruce Street, St. Louis, MO 63103, USA; Tel.: 314-771-5765). Pure Gram-positive bacterial cultures of *Listeria monocytogenes* (DSMZ 27575) and *Staphylococcus aureus* (DSMZ 12463) were obtained from the Institute of Technology of Agricultural Products, ELGO-DEMETER, Lykovryssi, Greece.

### 2.2. Preparation of Carvacrol Microemulsions and Nanoemulsions

In Table 1, the materials and quantities used for the preparation of carvacrol microemulsions and carvacrol nanoemulsions as well as their code names are listed.

The microemulsion containing 2.5% carvacrol (MC) was prepared by the self-emulsification method [23]. The appropriate amounts (see Table 1) of distilled water, Tween-80 as surfactant, and ethanol as co-surfactant were mixed in the beaker. Then, 2.5 mL of carvacrol was added under constant stirring (1000 rpm), and the transparent MC was obtained (see Table 1).

The microemulsion containing 2.5% carvacrol and diluted chitosan in the aqueous phase (MCC) was prepared similarly to the MC; however, acetic acid was first employed to dissolve CS in a 1:1 ratio before incorporating it into the aqueous phase (see Table 1).

The nanoemulsion containing 2.5% carvacrol (NC) coating was prepared by mixing the appropriate amounts (see Table 1) of distilled water, lecithin, casein, and carvacrol by using a high-speed ultrasonic homogenizer at 15,000 rpm and at ambient temperature for 20 min (see Appendix A).

### 2.3. Characterization of Obtained Carvacrol-Based Microemulsion and Nanoemulsions Dynamic Light Scattering (DLS) Measurements and Contact Angle Measurements

The DLS measurements were carried out using a high-performance two-angle particle and molecular size analyzer (Zetasizer Nano ZS, Malvern, Manchester M1 4ET, UK). The apparatus operated at a wavelength of 633 nm and was equipped with a helium-neon laser at the standard angle of 173°. For the experiments, a glass cuvette with a square aperture was used. The measurements were conducted to evaluate the hydrodynamic diameter and the polydispersity index of the solutions, at 37 °C. The hydrodynamic radius R_H_ (m) calculation is based on the Einstein–Stokes equation:(1)RH=kBT6πηD
where k_B_ (J/K) is the Boltzmann constant, T (K) is the absolute temperature, η (N⋅s⋅m^−2^) is the solvent’s dynamic viscosity, and D (m^2^⋅s^−1^) is the diffusion coefficient. The concentration of the solutions was approximately 0.1 wt%.

The study on wetting characteristics of MC, MCC, and NC coating samples was carried out using a contact angle instrument (OCA 25, DataPhysics Instruments GmbH, Filderstadt, Germany). Solutions with specific concentrations were prepared and then spin-coated onto silicon wafers under specific conditions (3500 rpm for 30 s), resulting in films with a thickness of approximately 40–50 nanometers. The silicon wafers were initially treated with a piranha solution (sulfuric acid/hydrogen peroxide in a 3:1 ratio). During the measurements, 4 μL droplets of deionized water (DI water) were deposited at a consistent rate of 0.5 μL/s. Three separate measurements were conducted on different regions of the same wafer, and the average value was computed and presented in the obtained results. The deviation in all instances did not surpass ±2 degrees, demonstrating the consistency of the results attributable to uniform film deposition.

### 2.4. Total Antioxidant Activity of Carvacrol-Based Microemulsion and Nanoemulsions with DPPH (2,2-diphenyl-1-picrylhydrazyl) Assay

#### 2.4.1. Preparation of DPPH Free Radical Standard Solutions

A standard DPPH ethanolic solution was applied according to the method proposed by Krishnanad et al. [24]. Briefly, 0.0212 g of DPPH radical was dissolved in 250 mL of ethanol to obtain an ethanolic DPPH solution of 2.16 mM (mmol/L). The flask was subjected to vortex mixing under dark conditions. The resultant DPPH··solution exhibited a deep purple hue and had a neutral pH of 7.02 ± 0.01. To ensure stability, the solution was refrigerated for 2 h before use. Once the free radical was stable, appropriate dilutions were carried out to establish the calibration curve.

#### 2.4.2. Preparation of DPPH Free Radical Calibration Curve

To determine the relationship between absorbance and concentration of DPPH, a calibration curve was constructed. The initial standard ethanolic solution of DPPH was diluted with ethanol to obtain fie standard solutions with the concentration range from 0 to 60 mg/L (ppm). Next, the absorbance of these five standard DPPH ethanolic solutions was measured using a Jasco V-530 UV/VIS Spectrometer (V-530 Jasco, Champaign, IL, USA) at a λ_max_ of 517 nm [25], and a linear calibration curve was developed with the obtained experimental data according to the methodology described previously [26].

#### 2.4.3. Estimation of Concentration Required to Obtain 50% Antioxidant Activity (ΕC50) of MC, MCC, and NC Coatings

For the estimation of concentration required to obtain 50% antioxidant activity (EC50) of the obtained MC, MCC, and NC coatings as well as pure carvacrol, 20, 40, 60, 80, and 100 μL of each coating (three times each) were added in 2.8 mL of 30 ppm DPPH ethanolic solution and 0.2 mL of CH_3_COONa·3H_2_O buffer solution. For comparison, a blank solution with 2.8 mL of 30 ppm DPPH ethanolic solution and 0.2 mL of CH_3_COONa·3H_2_O buffer solution was prepared. Next, the absorbance at 517 nm after 1 h, which was enough time to reach steady state conditions, was measured for all film coatings $A_{sample}^{517}$ as well as for blank coatings $A_{0}^{517}$. The antioxidant activity of all coatings was predicted indirectly by the calculated % remaining DPPH^•^ at steady state using the following equation:(2)$$\%\;\;\mathrm{remaining}{\mathrm{DPPH}}^{•}\mathrm{at}\mathrm{steady}\mathrm{state}=\frac{A_{0}^{517}-A_{sample}^{517}}{A_{0}^{517}}\times 100$$

The lower the % remaining DPPH^•^ at steady state, the reversed proportionally higher antioxidant activity of the antioxidant material [27]. Next, the calculated values of the % antioxidant activity of films were plotted as a function of the film quantity used, and the linear equation from the obtained plot was calculated. From the obtained linear equations for each coating, the EC50 value is calculated.

### 2.5. Antibacterial Activity Test of Carvacrol-Based Microemulsion and Nanoemulsions

#### 2.5.1. Inhibition Zone Tests

The antimicrobial efficacy of various carvacrol formulations was investigated using the well diffusion method against two critical foodborne pathogenic bacteria, specifically *Staphylococcus aureus* (DSMZ 12463) and *Listeria monocytogenes* (DSMZ 27575). These bacterial strains were sourced from the Institute of Technology of Agricultural Products, ELGO DEMETER, situated in Lykovryssi, Greece. To begin the study, the bacterial strains were cultured in Müeller–Hinton broth and incubated at 37 °C for a duration of 24 h. This enabled bacterial growth and led to a bacterial concentration that ranged between 10^7^ and 10^8^ colony-forming units per milliliter (CFU mL^−1^). Thereafter, the cultured bacteria were uniformly spread on Müller–Hinton agar plates using a sterile swab. The plates were rotated at intervals of 60 degrees to guarantee an even distribution of bacterial colonies. A number of 6mm wells were created on the agar surface of each plate using a cork borer that had been sterilized by dipping in alcohol and subsequently flaming. These wells were filled with 100 μL of the carvacrol suspensions being studied, including the free carvacrol (FC) sample used as control, MC, MCC, and NC coatings. After the wells had been filled, the agar plates were incubated at 37 °C overnight. Following the incubation period, the diameters of the clear zones around each well were measured using calipers to evaluate the antimicrobial activity of the different carvacrol formulations. The clear zones represented the area where bacterial growth was effectively inhibited by the respective suspensions. All experimental steps were performed in triplicate to ensure consistency and reliability in the acquired data. By using this methodology, we aimed to offer a comprehensive insight into the antimicrobial capabilities of different carvacrol formulations against *Staphylococcus aureus* and *Listeria monocytogenes*.

#### 2.5.2. Minimum Inhibition Concentration (MIC) Tests

The MIC serves as the lowest concentration of the antimicrobial agent at which the visible growth of the microorganism is inhibited, primarily indicating the bacteriostatic effect of the agents without specific data on the microbial population condition. Pure cultures of *Listeria monocytogenes* (LM) and *Staphylococcus aureus* (SA) were cultivated in Müeller–Hinton broth to concentrations approximating 10^−6^ CFU/mL. Four types of antimicrobial agents were evaluated: MC, MCC, NC, and FC. These agents were tested at multiple concentrations, including 1000, 500, 250, 125, 62.5, 31.25, and 15.625 μg/mL [28]. Utilizing the Macro dilution method, serial decimal dilutions of the microbial cultures were conducted, and the agents were added at each of these concentrations for evaluation. Following thorough mixing via a vortex mixer, the tubes were incubated at 37 °C for 24 h. Control tubes, which contained only microbial cultures without any antimicrobial agents, were also maintained and assessed at a concentration of 250 μg/mL. Turbidity in the tubes was observed as an indicator of microbial growth, and additional validation was performed through culture and colony counting techniques. All tests were conducted in triplicate to ensure the reliability and repeatability of the observed results.

### 2.6. Application of Carvacrol-Based Microemulsions and Nanoemulsion as Active Coatings in Fresh Minced Pork Meat

Fresh minced pork meat was provided by a local meat processing plant (Aifantis Company-Aifantis Group—Head Quarters, Acheloos Bridge, Agrinio, Greece 30100) and immediately transported to the laboratory. For each of the five treatment categories (Uncoated, FC, MC, MCC, and NC), 30 g of the minced meat was weighed. Each meat coating was then dipped into 25 mL of the appropriate treatment solution to ensure a uniform concentration of coated solution across all coated meat samples (see Appendix A). Following the dipping process, the meat was wrapped in a specialized membrane and stored at a temperature of 4 °C for a period of nine days. Physicochemical analyses such as pH measurements, *L*a*b** analysis and lipid oxidation analyses, microbiological analyses, and sensory analysis were conducted at three-day intervals throughout the storage duration to assess the efficacy of the applied treatments in meat preservation.

### 2.7. Physicochemical Properties of Pork Minced Meat

Chemical parameters including pH, color value (*L*a*b**), and thiobarbituric acid reactive substances (TBARs) were analyzed in pork minced meat coatings at specific time intervals: Days 0, 3, 6, and 9.

#### 2.7.1. pH Analysis

The pH values of the pork minced meat coatings were measured using a portable pH meter fitted with a penetration electrode and a temperature sensor (pH-Star, Matthäus GmbH, Poettmes, Germany). Prior to each set of measurements, the pH meter was calibrated using pH standard solutions of 4.0 and 7.0 and temperature-adjusted to match the meat coating temperature of 4 °C. The entire study was conducted in triplicate, and for each treatment group, ten separate pH readings were taken to ensure accuracy and reliability, as per the methods [29]. Overall, all coatings displayed an increase in pH over the 9-day analysis period.

#### 2.7.2. *L*a*b** Analysis

The alterations in the CIELAB color parameters (*L**, *a**, and *b**) of pork minced meat over a period of 9 days under refrigerated storage were assessed using an LS171 colorimeter from Linshang Company (Shenzhen, China) (see Appendix A). Prior to conducting the measurements, the colorimeter was calibrated with a white standard plate in accordance with the methods cited in Kang et al. (2019). Color evaluations were conducted directly on the surface of the minced meat coatings, with each treatment group comprising three separate portions. For each of these portions, nine discrete readings were taken to capture a robust assessment of the color. The total color differences (Δ*E*) were calculated using the equation [30]:$$\Delta E=\sqrt{{\left(L^{*}-L_{0}^{*}\right)}^{2}+{\left(a^{*}-a_{0}^{*}\right)}^{2}+{\left(b^{*}-b_{0}^{*}\right)}^{2}}$$

In this equation, *L**_0_, *a**_0_, *b**_0_ denote the initial color parameters of the pork minced meat at Day 0 post-treatment. *L**, *a**, *b** and represent the respective color parameters at different time points during the 9-day refrigerated storage at 4 °C.

#### 2.7.3. Lipid Oxidation TBARS

The level of lipid oxidation in meat coatings was measured using the thiobarbituric acid reactive substances (TBARS) method, based on Tarladgis et al. (1960) with minor alterations [31]. A 10 g coating of meat was combined with 97.5 mL of distilled water and 2.5 mL of 4 N HCl in a 500 mL flask. This mixture was steam distillated for roughly 20 min to yield 50 mL of distillate using a steam distillation apparatus (see Appendix A). From this distillate, a 5 mL coating was mixed with an equal volume of 0.02 M 2-thiobarbituric acid solution in a 15 mL tube. The tube was sealed and heated in a boiling water bath for approximately 35 min to form the TBA-Malondialdehyde (TBA-MA) chromogen. After cooling, the absorbance was measured at 532 nm using a DU-8 spectrophotometer. The conversion of TBA-MA absorbance readings to TBA numbers was carried out using a conversion factor of 7.8, as determined by the 1,1,3,3-tetramethoxypropane standard.

### 2.8. Microbiological Analyses of Stored Fresh Pork Minced Meat

Fresh pork minced meat coatings were evaluated for mesophilic bacteria, lactic acid bacteria (LAB), and psychotrophic bacteria on Days 0, 3, 6, and 9 of storage at 4 °C. An aseptic transfer of a 10 g pork sample was conducted into a stomacher bag, which was subsequently homogenized with 90 mL of peptone water. This was followed by a serial dilution (1:10 ratio). For the quantification of mesophilic bacteria, 0.1 mL from the diluted samples was spread onto PCA agar plates using the spread plate method and then incubated at 37 °C for 48 h. Psychotrophic bacteria were enumerated using the pour plate method; 1 mL of the sample was added to the plate, covered twice with the specific agar, and then incubated at 4 °C. For LAB determination, the MRS agar was employed with the pour plate method, and samples were incubated at 30 °C for 72 h [32]. All results were presented in terms of log CFU/g.

### 2.9. Sensory Evaluation of Stored Fresh Minced Pork Meat

The sensory attributes of the minced meat samples were evaluated throughout the storage period by a trained panel of 15 members, who have a substantial background in meat and sensory evaluation. The assessment was carried out using a 9-point hedonic scale to rate various attributes such as color, appearance, odor, texture, and overall acceptability, with 9 being “like extremely” and 1 being “dislike extremely”. A score of 5 served as the lower limit for acceptability, as outlined in the methodology proposed by [33]. Each sample was assessed in triplicate for each replicate, resulting in a total of six evaluations per treatment group.

### 2.10. Statistical Analysis

In this study, each experimental condition was replicated three times to validate the consistency and accuracy of results. Statistical analyses were performed using IBM SPSS version 22 (IBM Armonk, NY, USA), employing non-parametric tests suited for the data characteristics, with a focus on the Kruskal–Wallis H test for comparing multiple independent samples. A significance level of *p* = 0.05 was established as the threshold for significance in these analyses.

## 3. Results

### 3.1. Characterization of Carvacrol Micro and Nanoemulsion Coatings

Figure 1 presents the particle size distribution plots obtained from the DLS experiments for MC, MCC, and NC samples.

Lines (1), (2), and (3) in Figure 1 show the particle size distribution of MC, MCC, and NC coatings correspondingly. The calculated average particle size hydrodynamic diameter for MC coating is approximately 10 nm, for MCC coating approximately ~93 nm, and for NC coating approximately ~21 nm.

The increase in CS NE droplet size has been reported in several studies. Increasing the initial content of essential oils like carvacrol in CS nanoparticles leads to larger droplet sizes, likely due to reduced stability of the nanoparticle dispersion in water and interactions between CS’s amino groups and carvacrol. These interactions result in a less positively charged surface, contributing to droplet enlargement [34,35,36].

In Figure 2, the images of water contact angles of the obtained (1) MC, (2) MCC, and (3) NC samples are shown for comparison.

Wettability is an important aspect in the practical applications of such active coatings. To study the wettability of the samples, the surface water contact angles of the films were measured. The contact angle values are shown in Figure 2. Normally, a contact angle <90° is hydrophilic, while that >90° is hydrophobic. The results confirm that all the films are hydrophilic. The contact angles were found to be 7.5°, 13.1°, and 33.5° for MC, MCC, and NC samples, respectively. Thus, all obtained coatings are hydrophilic. When CS was added to the MCC coating, the hydrophilicity slightly decreased in comparison to the hydrophilicity of the MC coating without CS. Moreover, the NC coating sample exhibited the lowest hydrophilicity or the highest hydrophobicity.

### 3.2. Antioxidant Activity

The variations of obtained antioxidant activity values as a function of MC, MCC, and NC quantity used were plotted in Figure 3, and the data were fitted to linear equations (see Table 2).

These equations allowed us to derive the effective concentrations (EC50) required to achieve 50% of the maximum efficacy for each treatment.

At first glance, the obtained EC50 values for MC, MCC, and NC samples are much lower than the obtained EC50 value of FC. This fact validates both microemulsion and nanoemulsion technologies as technologies that minimize the quantity of active agents used and maximize the obtained bioactivity. Thus, for FC, we obtained a substantially high EC50 value of 422.27 mg/mL while for MC, MCC, and NC samples, the obtained EC50 values were 118.60 mg/mL, 110.64 mg/mL, and 97.56 mg/mL, respectively. The lowest EC50 values are obtained for the NC sample, implying the superiority of nanoemulsion technology over microemulsion ones. In advance, the obtained lower EC50 value of the MCC sample as compared to the obtained EC50 value of the MC sample suggests the enhancement of the antioxidant activity of microemulsion by the presence of CS chains in the external microemulsion’s aquatic phase [37]. The lower EC50 values for NC and MCC suggest enhanced efficacy compared to FC, echoing findings from previous studies on other types of treatments [38,39]. The mechanism underlying the efficacy of these treatments is not completely understood but is believed to involve complex interactions at the molecular level that enhance their effectiveness. These EC50 values serve as critical indicators of the efficiency and potential applications of these treatments in the food industry, particularly in meat preservation. The results of antioxidant effectiveness of nanoemulsions are in line with previous studies [40,41], further supporting the potential utility of these formulations.

### 3.3. Antibacterial Activity Test of Carvacrol-Based Microemulsion and Nanoemulsions

#### 3.3.1. MIC

In Table 3, the calculated minimum inhibitory concentration (MIC) values of MC, MCC, and NC samples as well as the FC sample against two bacterial strains, *Staphylococcus aureus* and *Listeria monocytogenes*, are listed for comparison. Representative images of the obtained diffusion zone of MC, MCC, NC, and FC samples are shown in Appendix A.

The MIC values represent the lowest concentration of carvacrol required to completely inhibit bacterial growth and are expressed in μg/mL. The MIC values listed in Table 3 reveal that all tested carvacrol microemulsions and nanoemulsions effectively inhibited the growth of both *Staphylococcus aureus* and *Listeria monocytogenes* at notably low concentrations, with MIC values ranging from 62.5 to 500 μg/mL. Specifically, the NC sample exhibited the lowest MIC values, at 62.5 μg/mL for both bacterial strains. The inclusion of CS in the carvacrol microemulsion enhanced its antimicrobial activity, with MIC values at 125 μg/mL for both strains (see Appendix A). Even though these values agree with earlier studies [28,42], the explanation provided in these references does not fit our case. According to the reported explanation, the reason for nanoemulsions’ higher effectiveness as compared to the relevant effectiveness of microemulsions is that they can deliver larger amounts of antimicrobial agents to bacterial cells due to smaller droplet size and higher surface area [43]. It is obvious from Figure 1 that, despite the fact the droplet size of NC was by far smaller compared to the size of the MCC material, the droplet size of the NC was close enough but larger compared to the size of the MC material. Thus, this explanation could not be sufficiently supported in our study. A possible explanation could be the different composition of such NC, MC, and MCC nano- and microemulsions, which is attributed to the NC’s lowest hydrophilicity. Overall, the data support the notion that carvacrol nanoemulsions could serve as a promising antimicrobial agent in food preservation applications, given their potent inhibitory effects on both *Staphylococcus aureus* and *Listeria monocytogenes*.

#### 3.3.2. Well Diffusion Zone

In Table 4 the calculated inhibition zones of MC, MCC, and NC samples as well as the FC sample for *Listeria monocytogenes* and *Staphylococcus aureus* are listed for comparison. Representative images with the results of MIC of MC, MCC, NC, and FC samples are shown in Appendix A.

As obtained from the list in Table 4 of inhibition zone values, the NC sample demonstrated the most potent antibacterial activity, with a minimum inhibition zone measuring 20 mm for *Listeria monocytogenes* and 21 mm for *Staphylococcus aureus*. Following closely, the MCC coating achieved inhibition zones of 10.1 mm and 11.2 mm for *Listeria monocytogenes* and *Staphylococcus aureus*, respectively. The standard MC coating had slightly lower efficacy, with zones measuring 8.3 mm and 9.3 mm for the two bacteria. Notably, the control groups showed no inhibition zones, affirming the antibacterial potential of the carvacrol microemulsions and nanoemulsion (see Appendix A). These findings indicate that carvacrol, particularly in nanoemulsion form, could be a potent candidate for antibacterial applications due to its increased surface area, improved solubility and stability, enhanced penetration into microbial cells, and the ability for targeted delivery of the antimicrobial agent [28].

### 3.4. Physicochemical Properties of Pork Minced Meat

#### 3.4.1. pH Analysis

In Table 5, the obtained pH values for all treatments used and for the 9-day examined period are listed for comparison.

It is obvious from Table 5 values that the coefficient of variation (% CV) is far below the value of 5% for all cases. Thus, the significance of the reliability of all data is very high. Moreover, according to the statistical analysis, which is presented in Appendix A Appendix A, at the end of day 9, the differences between pH values of the uncoated sample and the MC sample that did not contain chitosan were also significant. The initial pH levels of the pork samples varied between 5.29 and 5.75, which aligns with previously published data by other researchers [44]. At the onset of the experiment (Day 0), the pH levels in the treatment categories FC, MC, MCC, and NC were generally lower compared to the uncoated control group. The low MCC values can likely be attributed to the acidic nature of the solvents used in some coating processes [44].

Over the storage period, there was a consistent upward trend in pH across all treatment and control groups. By the 9th day, pH values spanned from 6.05 to 6.74. Of all the treatments, MCC and MC were particularly effective in modulating pH changes over time, as evidenced by their elevated pH levels of 6.55 and 6.74, respectively, by the ninth day. From Day 3 to Day 6, the pH difference between MCC and MC is considered statistically significant, while on Day 9 the pH difference between NC and the uncoated sample is also considered statistically significant (See Appendix A).

This rise in pH over time is commonly linked to the growth of spoilage microbes that break down proteins and generate alkaline compounds. The slower pace of this increase in the treated coatings suggests that the coatings played a role in mitigating bacterial activity. This could be due to the barrier formed by the coatings, which reduces exposure to air and thus limits microbial proliferation [35,44,45].

#### 3.4.2. Lipid Oxidation

In Figure 4, the calculated TBARS mean values of an uncoated pork mince sample as well as of pork minced meat samples coated with FC, MC, MCC, and NC are plotted as a function of days of storage.

The obtained TBARS values for Day 0 for the pork minced meat were in the range of 0.38 to 0.43 mg MDA/kg, dependent on the coating type. Such values are in line with expectations for freshly processed pork minced meat [46].

For the uncoated pork minced meat sample, the obtained TBARS values were recorded at 0.655, 0.814, and 1.102 mg MDA/Kg meat on Days 3, 6, and 9, respectively. In comparison, the obtained TBARS values for coated samples with the free carvacrol (FC) were observed to be 0.647, 0.723, and 1.032 mg MDA/Kg meat over the same intervals.

Regarding the meat samples coated with MC, the obtained TBARS values were 0.611, 0.767, and 1.011 mg MDA/Kg meat on Days 3, 6, and 9, respectively. The MCC-coated samples exhibited TBARS values of 0.577, 0.699, and 0.983 mg MDA/Kg meat on Days 3, 6, and 9. The most effective coating in terms of limiting lipid oxidation was NC, where the TBARS values were 0.564, 0.640, and 0.736 mg MDA/Kg meat on Days 3, 6, and 9, respectively.

MDA concentrations in meat samples do not have any regulatory thresholds. However, concentrations exceeding 0.5 mg/kg suggest some degree of oxidation, while levels above 1.0 mg/kg are considered potentially unsatisfactory in various research studies [47,48]

It can be noted that the pork minced meat coated with NC displayed the slowest rate of lipid oxidation over the observation period, which suggests the potential antioxidant properties of the nanoemulsion. Specifically, the percentage difference between NC and uncoated at Day 6 and 9 accounts for 21.4% and 33.3%, respectively. Meanwhile, the difference between NC and FC (control sample) was 11.5% and 28.7%. In comparison, the “Uncoated” and FC coatings exhibited a more rapid rate of lipid oxidation. From Day 6, the results of NC are statistically significant with uncoated and MCC (see Appendix A). The results are in line with several studies that conclude that encapsulation of essential oil inhibits lipid oxidation [44,48]

#### 3.4.3. *L*a*b** Analysis

In Table 6, the *L*a*b** values of uncoated pork mince sample as well as of pork minced meat samples coated with FC, MC, MCC, and NC at Day 0, Day 3, Day 6, and Day 9 of storage are listed for comparison.

Importantly, the observed discoloration in the pork is attributed to the accumulation of hydrogen peroxide produced by lactic acid bacteria during storage [35]. Among the samples, uncoated ones were the most susceptible to color degradation, experiencing a 34.6% decrease in the *L** values, which underscores their inherent vulnerability to color changes over time (see Appendix A). In stark contrast, samples coated with NC manifested the least decline in *L** values, showing a mere 2.4% drop. The initial robustness of these samples can be partly attributed to the casein component in the coating, as they boasted the highest *L** value on Day 0. On the other hand, free carvacrol-coated samples had a dramatic immediate effect on the coloration, evident from the initial *L** value but saw a 19.8% decline by Day 9. This suggests a less stable color profile in comparison to NC, despite its significant initial color-changing impact. Interestingly, MC coatings exhibited minimal initial impact on color, with an *L** value closest to that of the uncoated samples on Day 0. However, by Day 3, the carvacrol in the coating led to a nearly 10-unit increase in *L** values, totaling a 9.8% overall increase by Day 9. Conversely, MCC coatings showed only a moderate 3.6% change in *L** values but registered the steepest decline among the coated samples. In regards to *a** values (see Appendix A), free carvacrol was the most volatile, with a startling 114.5% increase, while MCC proved to be the most stable, altering by only 17.9%. For *b** values (see Appendix A), uncoated coatings displayed the most change, with a 38% increase, and MCC the least, with a 0.8% change (see Appendix A). In summary, the NC coating emerged as the most promising candidate for preserving color stability, highlighting the importance of effective coatings in mitigating hydrogen peroxide-induced discoloration in food preservation endeavors.

### 3.5. Results of Application of Carvacrol-Based Microemulsions and Nanoemulsions as Active Coatings for Pork Minced Meat Preservation

The effectiveness of examined formulations of carvacrol in inhibiting the growth of mesophilic bacteria in minced meat was assessed and presented in Figure 5 over a storage period of nine days at 4 °C.

The results show that the carvacrol microemulsions and carvacrol nanoemulsion inhibited the rapid growth of the mesophilic bacteria by 3 days. In particular, the sample with no coating experienced a substantial increase in mesophilic bacteria levels, while all carvacrol-based treatments maintained growth within acceptable limits throughout the nine-day period. Both FC and MC samples crossed the acceptance line of 6 log CFU by Day 9. In contrast, the MCC approached the threshold but remained beneath it. Most strikingly, the NC coating curtailed the microbial proliferation more effectively than free carvacrol, registering reductions by 1 log and 1.2 logs on Days 6 and 9, respectively. These results for the carvacrol nanoemulsion are considered statistically significant at the 5% level on Days 6 and 9.

Examining the effectiveness of carvacrol-formulated coatings on the growth of LAB bacteria in minced meat over a 9-day storage period at 4 °C, the results illustrate varied inhibitory impacts (see Figure 6).

All samples began with an identical LAB bacterial count of 1.9 log CFU. Without any coating, the LAB bacterial levels escalated, culminating at 9.1 log CFU by Day 9. In comparison, the sample containing free carvacrol, which served as the control, recorded a rise to 6.7 log CFU over the same duration. The MC exhibited a similar growth trend, settling at 6.5 log CFU on Day 9. Meanwhile, the MCC demonstrated more restraint in bacterial growth, ending at 6.1 log CFU. Most notably, the NC showcased a significant bacterial inhibitory effect, with the LAB bacterial count only reaching 4.8 log CFU by the ninth day, reflecting a log reduction compared to the control. When compared against the set acceptable quality level of 6 log CFU, only the nanoemulsion-coated sample consistently stayed below this threshold. Moreover, the inhibitory effects observed in the NC-coated samples are considered statistically significant (*p* < 0.05), especially when compared with the free carvacrol formulations.

Finally, examining the efficacy of carvacrol-formulated samples on psychotrophic bacteria in minced meat over 9 days at 4 °C, distinct patterns emerged (see Figure 7).

Beginning at 3.8 log CFU, the uncoated sample experienced a significant increase in the psychotrophic bacteria levels to 9.3 log CFU by Day 9. The free carvacrol escalated to 7.4 log CFU in the same integral. The MC exhibited a similar trajectory, reaching 7.2 log CFU by Day 9. The presence of CS in the MCC sample resulted in a more moderated increase, peaking at 6.8 log CFU by the end of the observation period.

Most notably, the NC demonstrated a suppressive effect, with bacterial counts modestly rising from the initial 3.8 log CFU to only 5.3 log CFU by Day 9. This translates to a significant 2-log reduction compared to the free carvacrol by Day 9. Furthermore, compared to the set acceptable quality limit of 6 log CFU, the NC remained consistently beneath this standard throughout the storage period, underpinning the superior antibacterial potency of the nanoemulsified carvacrol. This marked reduction by the nanoemulsion was also found to be statistically significant at the *p* < 0.05 level.

Overall, the results are in accordance with other studies that use CS and carvacrol or similar essential oils to preserve meat products and found significant log reduction using nanoemulsions [49,50,51].

### 3.6. Sensory Analysis

The sensory evaluations of minced pork meat treated with various carvacrol coatings are detailed in Table 7.

Even though many values of coefficient of variation (% CV) in Table 7 are greater than the commonly trusted 10%, the non-parametric hypothesis test of means equality indicated that such mean values of sensory tests were statistically different at a significance level *p* < 0.05. Thus, we can conclude the following: At the beginning of the storage period (Day 0), all the samples across the different coating groups registered high scores in terms of appearance, color, odor, texture, and overall acceptability. As the storage period extended, there was a noticeable trend where the scores for all sensory parameters and overall acceptability witnessed a decline. This decline in sensory scores could likely be attributed to factors such as alterations in meat consistency or the emergence of undesirable flavors or odors over time.

From Day 3, the MCC and NC-coated samples consistently exhibited the highest scores across all sensory attributes, beginning on Day 3 and continuing until the culmination of the observed storage period. Upon assessing the longevity of acceptability based on appearance, color, texture, and overall impression, it is evident that the uncoated and FC-coated minced pork meat samples maintained their acceptability until Day 9. In contrast, the MC, MCC, and NC-coated samples sustained their sensory acceptability through Day 9, with MCC-coated samples displaying marginally superior resilience. With respect to odor scores, the uncoated and FC-coated samples remained within acceptable bounds for 6 days, while the MC, MCC, and NC-coated samples retained their acceptability for a duration of up to 9 days (see Appendix A and Appendix A).

The data collated suggest that the introduction of specific carvacrol coatings, particularly MCC, can substantially elevate the sensory attributes of minced pork meat throughout the storage phase, potentially extending its desirability and shelf life by a few days. Such findings are in accordance with contemporary research in the meat processing sector [52]. This study concentrates on the influence of different carvacrol coatings on the sensory attributes of minced pork meat during its storage. A progressive approach would entail an expansive exploration into the sensory properties of carvacrol-coated pork when exposed to various environmental conditions or culinary methodologies.

## 4. Discussion

As it was shown in the results section, both microemulsion and nanoemulsion carvacrol coatings exhibited high performance in the preservation of fresh minced pork meat. Overall, NC coating exhibited the highest antioxidant activity according to obtained EC50 values, the highest antibacterial activity against *L. monocytogenes* and *S. aureus* according to obtained MIC values and well diffusion zone results, the lowest pH values increment during the minced pork meat preservation process, the lowest lipid oxidation during the minced pork meat preservation process according to obtained TBRAS values, the highest pork minced pork meat color stability during the preservation process, and the lowest mesophilic, LAB, and psychotrophic bacteria growth during the minced pork meat preservation process. The microbiological analysis of fresh minced pork meat stored at 4 °C shows that the uncoated pork meat reaches the upper microbiological limit (6 log CFU/g) for acceptable quality of foods according to the ICMSF [53]. Both MC and MCC coating samples succeeded in preserving the stored minced pork meat until the ninth day of storage. NC coating samples succeed in achieving much lower values of mesophilic, LAB, and psychotrophic bacteria growth as compared to the upper microbiological limit of 6 log CFU/g, and promise a preservation period of fresh minced pork meat over the nine days of storage. The physicochemical characterization of such novel MC, MCC, and NC coatings indicated that the calculated average droplet size hydrodynamic diameter is approximately ~10 nm for MC coating, approximately ~93 nm for MCC coating, and approximately ~21 nm for NC coating. Thus, the obtained size of carvacrol nanodroplets in the MC, MCC, and NC nanocoating does not offer a satisfactory explanation for the superiority of NC coating versus both MC and MCC coating samples in the preservation of fresh minced pork meat. A possible explanation could be the biocompatibility of NC coating with pork meat compared to the relevant MC and MCC coatings. The reason is that the biobased casein and lecithin, which are used as surfactants and co-surfactants, respectively, to obtain the NC coating nanostructure, are known as ideal biobased encapsulation agents for such essential oils and their derivatives because of their amphiphilic nature [43,44,45,46]. Compared to MC and MCC coating samples, the contact angle results of NC coating indicated the lowest hydrophilicity nature of this material, which is similar to casein’s amphiphilic nature. This lowest hydrophilicity or higher hydrophobicity of casein could be a logical explanation for the highest biocompatibility of such NC nanocoating with minced pork meat. Such compatibility supports the higher diffusion of NC inside the casein and the controlled release of carvacrol nanodroplets in minced pork meat.

## 5. Conclusions

The preparation, characterization, and application of carvacrol coatings were elaborated upon in the present study. Different emulsion forms, i.e., nanoemulsion with carvacrol NC, microemulsion with carvacrol MC, and microemulsion with carvacrol and chitosan MCC, were examined for their preservation capacity against pork meat spoilage after a 9-day storage period under 4 ± 1 °C temperature. As an overall conclusion, we could say that the NC form is superior to the other two forms of emulsion. Based on Figure 1 mean sizes, we could say that this happens not because of the higher external specific area of the droplets but because of the higher biocompatibility of the NC material compared to the relevant MC and MCC materials. More specifically, according to Figure 2, all samples are hydrophilic but the least hydrophilic was the NC material. This happens because of the amphiphilic nature of casein and lecithin, which were used as surfactants for the encapsulation of carvacrol. The more organophilic nature of the NC led to better incorporation and cooperation with carvacrol oil. In this direction was also the contribution of chitosan, which is antimicrobial by itself, but its contribution was lower than casein and lecithin. Apart from the antimicrobial activity, it is obvious from Figure 3 and EC50 values of Table 2 that carvacrol in the nanoemulsion variety exhibited higher antioxidant activity compared to the other two materials. It is also obvious that, for the MCC coating, chitosan enhances this activity compared to microemulsion without chitosan, i.e., MC material. The superiority of the NC material in antibacterial activity was confirmed by experimental measurements on *Staphylococcus aureus* and *Listeria monocytogenes* presented in Table 3. This result is supported by well diffusion assay measurements, statistically verified in Appendix A and tabulated in Table 4. Another indication that carvacrol serves the food preservation process better in nanoemulsion form NC than in microemulsion form MC or MCC was the stability of pH values presented in Table 5, where, according to Appendix A, after 9 days, the mean value of the NC sample was statistically equal with the relevant of sample MCC because of chitosan but still lower than that of this sample. Figure 4 presents the inhibition of lipid oxidation activity by all tested samples. It is obvious that at the end of the 3rd day, the NC material was distinguished from the other, and at the end of the 9th day, the difference was too high. Similarly to the above observations, chitosan offers some extra protection but not enough to overlap the activity of the NC material. The results are also supported statistically by Appendix A. According to ΔE values of Table 6, the more stable color parameter of the pork meat was that of NC preserved meat. Appendix A confirm this result statistically. Figure 5, Figure 6 and Figure 7 show that, at the end of day 9, the MCC and MC coatings reached and overlapped, respectively, the acceptance line of mesophilic, LAB, and psychotrophic bacteria concentration while the NC coating remained far below this line. Even though, from the overall MCC and MC lines, chitosan seems to contribute to the antibacterial activity of the microemulsion, the NC antibacterial capacity dominated during the whole experiment. Such results were confirmed statistically at a significance level of *p* < 0.05 by Appendix A, respectively. Finally, sensory tests showed that MCC and NC emulsions exhibited higher stability after 9 days storage. Results are presented in Table 7 and confirmed statistically by Appendix A values.

## Figures and Tables

**Figure 1 nanomaterials-13-03161-f001:**
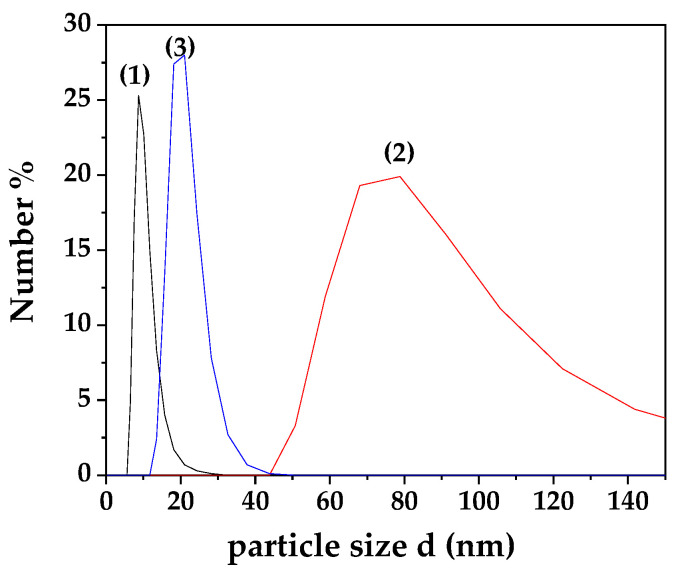
Particle size distribution plots obtained from DLS measurements of (1) MC, (2) MCC, and (3) NC coatings.

**Figure 2 nanomaterials-13-03161-f002:**
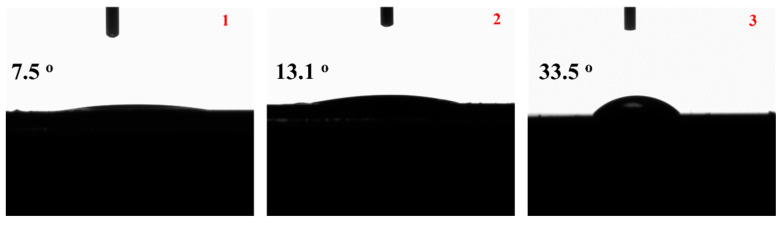
Images of water contact angles of (**1**) MC, (**2**) MCC, and (**3**) NC samples.

**Figure 3 nanomaterials-13-03161-f003:**
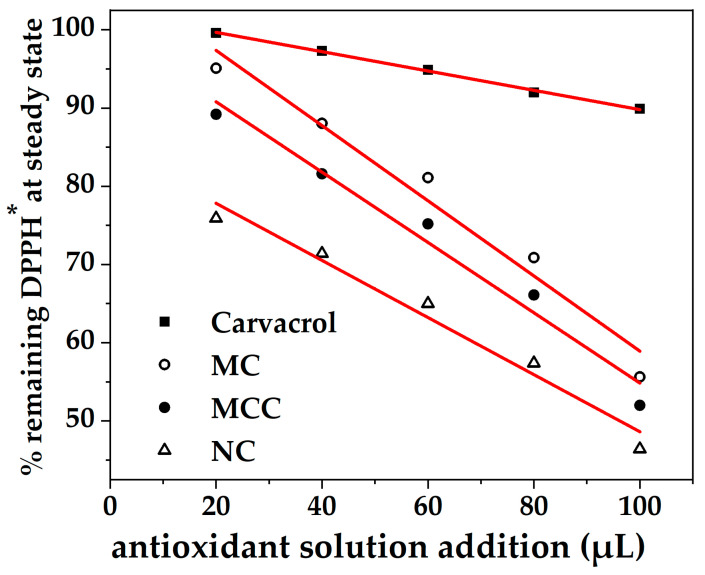
% Remaining DPPH^•^ at steady state, which is reversely proportional to the antioxidant efficacy of FC, MC, MCC, and NC edible coatings, versus the volume of essential oil used (μL).

**Figure 4 nanomaterials-13-03161-f004:**
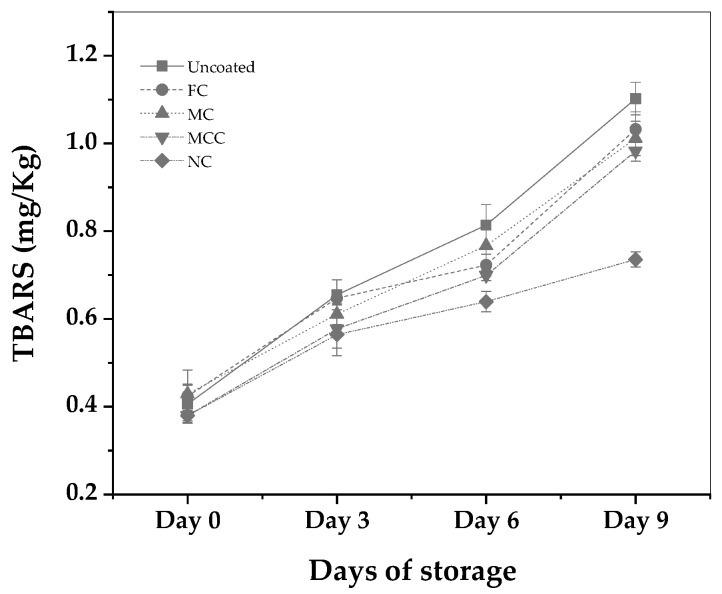
Calculated TBARS mean values of an uncoated pork mince sample as well as of pork minced meat samples coated with FC, MC, MCC, and NC as a function of days of storage. See also the statistical analysis results in Appendix A.

**Figure 5 nanomaterials-13-03161-f005:**
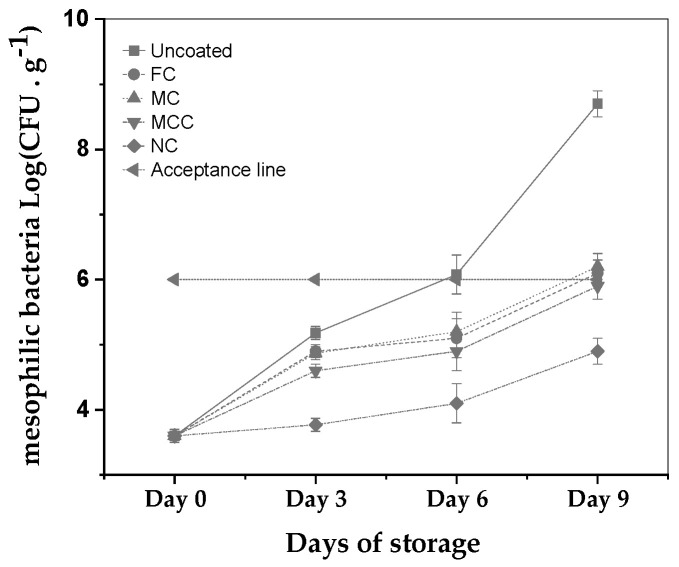
Mesophilic bacteria growth (Log CFU) during 9 days of storage at 4 °C for uncoated minced meat sample and minced meat samples coated with FC, MC, MCC, and NC. See also statistical analysis results in Appendix A.

**Figure 6 nanomaterials-13-03161-f006:**
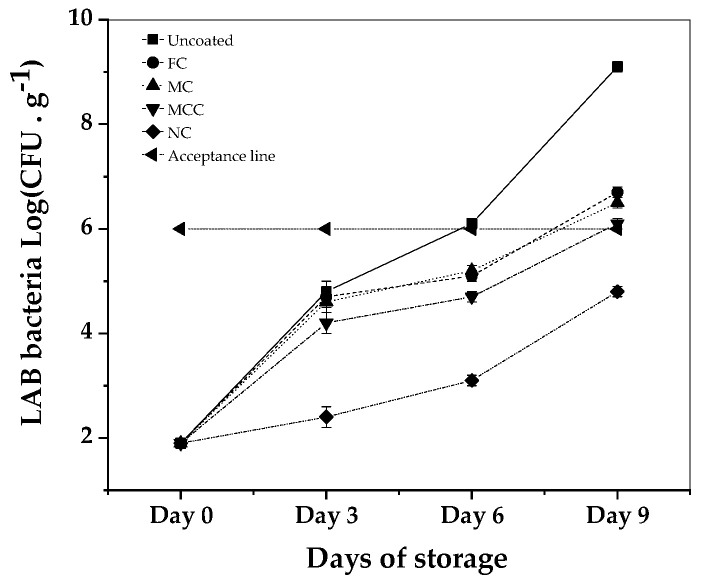
LAB Bacteria Growth (Log CFU) during 9 days of storage at 4 °C for uncoated minced meat sample and minced meat samples coated with FC, MC, MCC, and NC. See also statistical analysis results in Appendix A.

**Figure 7 nanomaterials-13-03161-f007:**
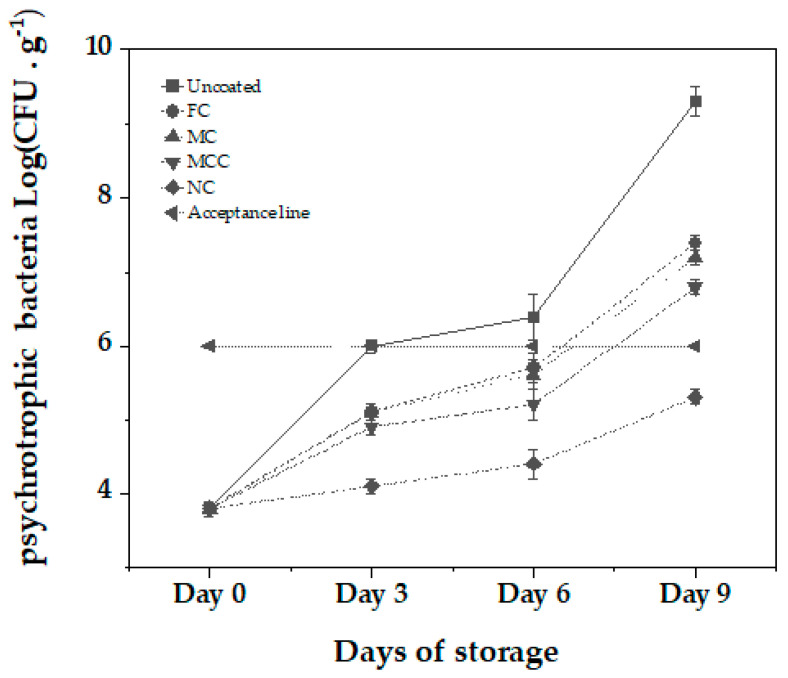
Psychotrophic bacteria growth (Log CFU) during 9 days of storage at 4 °C for uncoated minced meat sample and minced meat samples coated with FC, MC, MCC, and NC. See also statistical analysis results in Appendix A.

**Table 1 nanomaterials-13-03161-t001:** Composition of Carvacrol Microemulsions and Nanoemulsions.

Sample Description	Code Name	H_2_O (mL)	Tween-80 (mL)	EtOH (mL)	Lecithin (g)	Casein (g)	Carvacrol (mL)	Chitosan (g)	CH_3_COOH (mL)
Microemulsion with 2.5% carvacrol	MC	57.5	25.0	15.0	-	-	2.5	-	-
Microemulsion with 2.5% carvacrol and chitosan (MCC)	MCC	56.5	25.0	15.0	-	-	2.5	0.5	0.5
Nanoemulsion with 2.5% carvacrol	NC	97.5	-	-	0.5	4.0	2.5	-	-

**Table 2 nanomaterials-13-03161-t002:** Calculated EC50 values as well as slope, intercept, and R^2^ values of fitted linear equations for FC, MC, MCC, and NC coatings.

Sample Code Name	Slope	Intercept	R^2^	ΕC50 (μL)
FC	−0.12	102.15	0.998	422.27
MC	−0.48	106.98	0.978	118.57
MCC	−0.45	99.79	0.978	110.64
NC	−0.36	85.12	0.978	97.56

**Table 3 nanomaterials-13-03161-t003:** Minimum inhibitory concentrations (MIC) of carvacrol microemulsions and nanoemulsion against Staphylococcus aureus and Listeria monocytogenes.

Treatment Group	*Staphylococcus aureus*	*Listeria monocytogenes*
	MIC (μg/mL)	MIC (μg/mL)
FC	<500	<500
MC	<250	<250
MCC	<125	<125
NC	<62.5	<62.5

**Table 4 nanomaterials-13-03161-t004:** Inhibition zones of MC, MCC, NC samples as well as FC samples against *Listeria monocytogenes* and *Staphylococcus aureus* in well diffusion assay.

Treatment Group	Bacteria Tested	Minimum Inhibition Zone (mm)
FC Control	*Listeria monocytogenes*	2.2 (±0.15)
FC Control	*Staphylococcus aureus*	2.3 (±0.10)
MC	*Listeria monocytogenes*	8.3 (±0.50)
MCC	*Listeria monocytogenes*	10.1 (±0.70)
NC	*Listeria monocytogenes*	20.0 (±1.00) *
MC	*Staphylococcus aureus*	9.3 (±0.50)
MCC	*Staphylococcus aureus*	11.2 (±0.70)
NC	*Staphylococcus aureus*	21.0 (±1.00) *

SStatistical significance (*) determined by non-parametric test: *p* < 0.05. (see Appendix A).

**Table 5 nanomaterials-13-03161-t005:** pH Evolution in pork samples uncoated and treated with MC, MCC, NC, and FC coatings over a 9-day period. See also Appendix A with pH statistical analysis.

Treatment	Day 0	CV (%)	Day 3	CV (%)	Day 6	CV (%)	Day 9	CV (%)
Uncoated	5.75 (±0.02)	0.35	5.87 (±0.02)	0.34	6.55 (±0.02)	0.31	6.85 (±0.02)	0.29
FC	5.68 (±0.01)	0.18	5.80(±0.02)	0.34	6.42 (±0.02)	0.31	6.74 (±0.01)	0.15
MC	5.58 (±0.03)	0.54	5.93 (±0.02)	0.34	6.12 (±0.03)	0.49	6.45 (±0.01)	0.16
MCC	5.29 (±0.02)	0.38	5.32 (±0.01)	0.19	6.11 (±0.02)	0.33	6.15 (±0.03)	0.49
NC	5.75 (±0.03)	0.52	5.51 (±0.02)	0.36	5.78 (±0.03)	0.52	6.05 (±0.02)	0.33

Statistical significance assumed for non-parametric test: *p* < 0.05. (see Appendix A).

**Table 6 nanomaterials-13-03161-t006:** *L*a*b** values of the uncoated pork mince sample as well as of pork minced meat samples coated with FC, MC, MCC, and NC at Day 0, Day 3, Day 6, and Day 9 of storage.

Color Parameter	Storage (Day)	Uncoated	FC	MC	MCC	NC
*L**	0	64.76 ± 2.97 ^Aa^	63.46 ± 3.22 ^Aa^	48.72 ± 1.06 ^Aa^	52.27 ± 1.05 ^Aa^	74.22 ± 2.17 ^Aa^
	3	58.81 ± 0.20 ^Ab^	58.02 ± 0.10 ^Ba^	54.16 ± 0.55 ^Ba^	48.66 ± 0.96 ^Aa^	72.39 ± 1.81 ^Aa^
	6	52.86 ± 3.08 ^Bb^	52.47 ± 1.56 ^Cb^	60.26 ± 3.04 ^Cb^	45.27 ± 2.06 ^Aa^	70.56 ± 5.32 ^Aa^
	9	42.36 ± 2.88 ^Cd^	50.87 ± 2.74 ^Dc^	58.39 ± 2.84 ^Cc^	42.67 ± 1.73 ^Aa^	68.47 ± 4.11 ^Aa^
*a**	0	15.94 ± 0.53 ^Aa^	4.89 ± 1.03 ^Ec^	4.87 ± 2.31 ^Aa^	5.89 ± 2.37 ^Aa^	4.47 ± 3.54 ^Aa^
	3	12.88 ± 0.10 ^Aa^	5.31 ± 0.11 ^Aa^	6.07 ± 0.11 ^Aa^	1.71 ± 1.52 ^Aa^	5.75 ± 1.12 ^Aa^
	6	9.82 ± 1.97 ^Ab^	5.73 ± 0.86 ^Aa^	7.25 ± 2.23 ^Aa^	1.61 ± 1.52 ^Aa^	7.03 ± 3.94 ^Aa^
	9	7.80 ± 1.31 ^Ba^	5.63 ± 1.05 ^Aa^	6.88 ± 1.67 ^Ba^	1.45 ± 1.28 ^Aba^	6.29 ± 2.89 ^Aa^
*b**	0	3.87 ± 4.95 ^Ba^	7.15 ± 3.73 ^Da^	6.82 ± 2.07 ^Ba^	6.76 ± 1.50 ^Aa^	7.43 ± 1.80 ^Aa^
	3	4.74 ± 1.79 ^Da^	5.34 ± 0.07 ^CDa^	4.15 ± 2.82 ^Bb^	3.55 ± 0.90 ^Ab^	7.39 ± 1.23 ^Ab^
	6	5.43 ± 3.79 ^Ca^	4.86 ± 2.32 ^BCb^	3.98 ± 2.76 ^Bc^	3.45 ± 0.90 ^Ac^	7.29 ± 0.79 ^Ac^
	9	5.34 ± 1.36 ^Ca^	4.37 ± 1.29 ^Bb^	3.27 ± 2.02 ^ABc^	3.25 ± 0.83 ^Ad^	6.93 ± 0.95 ^Acd^
Δ*E*	3	6.75 ± 2.62 ^Aa^	5.74 ± 1.04 ^Aa^	3.17 ± 1.93A ^a^	6.38 ± 2.16 ^Ca^	2.23 ± 2.02 ^Ca^
	6	7.02 ± 4.87 ^Ea^	5.91 ± 1.93 ^Db^	1.69 ± 2.67 ^Db^	3.30 ± 2.89 ^Cc^	2.40 ± 4.59 ^Cc^
	9	10.36 ± 4.18 ^Da^	5.64 ± 1.69 ^Cb^	8.71 ± 3.12 ^Cc^	9.94 ± 2.86 ^Bd^	2.96 ± 4.07 ^Bd^

^A–E^ Any means within the same column not sharing a common letter are significantly different at the *p* < 0.05 level according to the non-parametric test. ^a–d^ Any means within the same row not sharing a common letter are significantly different at the *p* < 0.05 level according to the non-parametric test. See also statistical analysis results in Appendix A.

**Table 7 nanomaterials-13-03161-t007:** Sensory analysis of examined carvacrol treatment on minced meat.

Sensory Parameter	Storage (Day)	Uncoated	CV (%)	FC	CV (%)	MC	CV (%)	MCC	CV (%)	NC	CV (%)
Appearance	Day 0	8.87 (±0.35) ^Aa^	3.9	8.80 (±0.41) ^Aa^	4.7	8.80 (±0.51)^Aa^	5.8	8.40 (±0.51) ^Aa^	6.1	8.67 (±0.49) ^Aa^	5.7
	Day 3	6.87 (±0.86) ^Aa^	12.5	7.07 (±0.46) ^Ad^	6.5	7.40 (±0.49) ^Aa^	6.6	7.40 (±0.51) ^Ac^	6.9	8.07 (±0.88) ^Ab^	10.9
	Day 6	4.40 (±0.46) ^Ab^	10.5	4.87 (±0.46) ^Ae^	9.4	5.80 (±0.49) ^Abs^	8.4	5.80 (±1.23) ^Ae^	21.2	7.07 (±1.32) ^Af^	18.7
	Day 9	1.80 (±0.52) ^Ab^	28.9	2.20 (±0.51) ^Ac^	23.2	4.53 (±1.23) ^Ad^	27.2	5.53 (±0.86) ^Ae^	15.6	5.00 (±0.52) ^Af^	10.4
Odor	Day 0	8.60 (±0.51) ^Aa^	5.9	7.80 (±0.87) ^Aa^	11.2	7.80 (±0.64) ^Aa^	8.2	8.00 (±0.64) ^Aa^	8.0	8.07 (±0.66) ^Aa^	8.2
	Day 3	8.20 (±0.86) ^Aa^	10.5	6.40 (±0.46) ^Ab^	7.2	7.60 (±0.87) ^Aa^	11.4	7.60 (±0.69) ^Aa^	9.1	8.27 (±0.66) ^Ac^	8.0
	Day 6	5.27 (±0.46) ^Aa^	8.7	5.3 (±0.93) ^Ab^	17.4	6.33 (±0.69) ^Ac^	10.9	6.60 (±0.73) ^Ad^	11.1	6.60 (±0.65) ^Ae^	9.8
	Day 9	1.53 (±0.52) ^Ab^	34.0	1.60 (±0.95) ^Ac^	59.4	2.67 (±0.73) ^Ad^	27.3	3.80 (±0.73) ^Ae^	19.2	5.80 (±0.73) ^Af^	12.6
Color	Day 0	8.53 (±0.52) ^Aa^	6.1	8.60 (±0.51) ^Aa^	5.9	8.60 (±0.70) ^Aa^	8.1	8.27 (±0.70) ^Aa^	8.5	8.67 (±0.49) ^Aa^	5.7
	Day 3	7.13 (±0.92) ^Ab^	12.9	7.33 (±0.98) ^Ac^	13.4	7.73 (±0.70) ^Ad^	9.1	7.73 (±0.70) ^Ae^	9.1	8.20 (±0.68) ^Af^	8.3
	Day 6	6.27 (±1.03) ^Ab^	16.4	6.20 (±0.77) ^Ab^	12.4	6.87 (±1.06) ^Aa^	15.4	7.33 (±0.90) ^Ab^	12.3	7.33 (±0.86) ^Aa^	11.7
	Day 9	3.20 (±0.94) ^Bb^	29.4	3.27 (±0.59) ^Ac^	18.0	5.20 (±0.68) ^Ad^	13.1	6.00 (±0.76) ^Ae^	12.7	6.53 (±1.19) ^Af^	18.2
Texture	Day 0	8.73 (±0.46) ^Aa^	5.3	8.60 (±0.51) ^Aa^	5.9	8.60 (±0.49) ^Aa^	5.7	8.67 (±0.46) ^Aa^	5.3	8.73 (±0.46) ^Aa^	5.3
	Day 3	7.60 (±0.51) ^Aa^	6.7	7.47 (±0.52) ^Aa^	7.0	7.73 (±0.70) ^Aa^	9.1	8.13 (±0.68) ^Aa^	8.4	8.00 (±0.80) ^Aa^	10.0
	Day 6	5.73 (±0.80) ^Bb^	14.0	4.87 (±0.83) ^Bc^	17.0	6.67 (±0.62) ^Ad^	9.3	7.47 (±0.68) ^Ae^	9.1	6.40 (±0.99) ^Bf^	15.5
	Day 9	5.47 (±0.52) ^Ca^	9.5	5.73 (±0.59) ^Ab^	10.3	4.53 (±0.52) ^Ab^	11.5	6.20 (±0.68) ^Ab^	11.0	5.60 (±0.74) ^Ac^	13.2
Overall	Day 0	8.67	-	8.40	-	8.22	-	8.47	-	8.51	-
	Day 3	7.40	-	6.93	-	7.58	-	8.18	-	8.18	-
	Day 6	5.31	-	5.47	-	6.33	-	7.00	-	7.16	-
	Day 9	2.18	-	2.36	-	4.13	-	5.11	-	5.78	-

^A–E^ Any means within the same column not sharing a common letter are significantly different at the *p* < 0.05 level according to the non-parametric test. ^a–d^ Any means within the same row not sharing a common letter are significantly different at the *p* < 0.05 level according to the non-parametric test. Statistical significance assumed by non-parametric test: *p* < 0.05. (see Appendix A).

## Data Availability

The datasets generated for this study are available on request to the corresponding author.

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
