# Peer review of "Carvacrol Microemulsion vs. Nanoemulsion as Novel Pork Minced Meat Active Coatings"

_nanomaterials, 2023, doi:10.3390/nano13243161_

Round 1

Reviewer 1 Report

Comments and Suggestions for Authors

This work investigated the carvacrol microemulsion vs nanoemulsion as novel pork minced meat active coatings. The author did a novel research with sufficient sample quantity. However, there are still some issues, so I don’t think it can be published.

1. The Composition of the carvacrol microemulsions and nanoemulsions are different. Are you sure that the effect is due to the difference in particle size?

2. Line 164. DPPH assay should be written with the full name.

3. Line 252, 507, 509. The analysis of color, b should be written by adding *.

4. What about the replicates of this experiment and the parallel determination of the data? The authors should give a more detailed description?

5. Line 514. In the color value, there is no difference in the a value from 0 days to 3 days in different treatment groups, whether the data processing is correct

Author Response

Thank you for your effort to improve our work.

Please find online the file with our response.

Reviewer 2 Report

Comments and Suggestions for Authors

The submitted article examined and compared carvacrol microemulsion (MC), carvacrol microemulsion busted with chitosan (MMC) and carvacrol nanoemulsions(NC) as active coatings on extending minced pork meat shelf-life at 4±1 ℃ for 9 days, and it proves that the carvacrol based nanoemulsion can be considered as novel antioxidant and antimicrobial active coating due to its demonstrated higher efficacy in all the examined tests performed. Overall, the article has a certain degree of innovation and novelty, nonetheless, there are opportunities for improvement in the following aspects:

1.      Introduction: Is it possible to describe in the introduction why carvacrol needs to be encapsulated, and what are the drawbacks if it is not encapsulated ? At the same time, are there too few introductions about microemulsion and adding chitosan into microemulsion?

2.      Section 2.4: I would like to know the origin of your DPPH measurement method, as I have never seen anyone measure it like this before.

3.      L209: The description of abbreviations in the article should be accurate, just like FC did not mention it earlier, if it suddenly appears here, readers will not be able to distinguish which substance it is.

4.      Figure 2: The abscissa of Figure 2 is not very accurate, and it can be named as Solution addition amount (μL), rather than μL and attention should be paid to spelling errors in the annotations, meanwhile, I also want to know why the DPPH decreases with the increase of essential oil content.

5.      L387, L410: There is an error in the name expression of supplementary figures in L389 and L410. L389 should be MIC and L410 should be a diffusion zone.

6.      Section 3.3.1: When measuring MIC, you wrote in the method that you selected Staphylococcus aureus, Listeria monocytogenes, and Escherichia coli, but why did you not have data on Escherichia coli during the analysis.

7.      L402: It is explained that NC is better than MC because of its smaller droplet size and higher surface, however, in the determination of emulsion particle size in the previous article, the particle size of microemulsion is smaller than that of nanoemulsion , is this explanation accurate?

8.      Section 3.4.1: Why is the pH change of pork mince coated with MC and MCC close to that of uncoated pork mince on the ninth day, even reaching the limit of spoiled meat ?

9.      Conclusion: This article is about the comparison of carvacrol microelement (MC), carvacrol microelement boosted with chitosan (MMC) and carvacrol nanoemulsions (NC) as active coatings, why is there only a summary of NC in the conclusion.

10.   It is necessary to pay attention to the aesthetics of the tables and the clarity of the images in the article, in order to improve its readability and professionalism.

Author Response

(The authors gave the same response as above.)

Reviewer 3 Report

Comments and Suggestions for Authors

P1, L19: I suppose that 4 oC sholud have been 4°C, but it got distorted when you copied the text into the template. Please correct the symbol.

Abstract: What does the abbreviation NC mean? Please add the full name when mentioning the abbreviation for the first time.

P1, L35: What are safety foods? Please rephrase. Also, why is in "Food Technology" each word capitalized?

P3, L 106-111: Ethanol, sodium caseinate, lecithin and soy bean should not be capitalized.

P4, L148-150: Please add measuring units for each symbol explained in the text.

P4, L183: Bacterial names should be written in itallic.

P4, L181 - 202: Why is Carvacrol capitalized? Please correct throughout the manuscript.

P6, L251 - 266: L*, a* and b* should be written in itallic. Please correct throughout the manuscript.

Tables 5 and 7: Statistical significance data is missing.

What is the infuence of particle size on the antioxidant capacity, antibacterial activity and sensory properties? Please discuss in detail.

Please revise the Conclusions according to the statistical analysis performed as mentioned in the comment above.

P7, L23: You mention that entirely random design was employed. What does that mean? Which software, which factors at how many levels were used? If you mention experimental design then those things should be defined. It seems to me that there was no experimental design present here, you only randomly chose 3 formulations.

In Table 1, you list 3 samples. However, in the R&D section, there are 4 samples shown in tables and graphs. Furthermore,  in Table 5, there are 5 samples. Please list the exact number of samples and the sample names consistently throughout the manuscript.

P6, L306-311: You only had 3 (4 or 5?) samples for statistical analyses. Therefore, I recon that the data obtabtained are not normally distributed, so t-test and ANOVA as parametric statistical tests cannot be applied because they can result in false positives or incorrect values since the number of samples is 5. For parametrics, minimal n should be 6. In your case, the statistical analysis should be corrected. First, test the normality of the distribution (and list the test you used clearly in the M&M). Then, if the numbers are distributed normally, you can use ANOVA or t-test. If not, you can only use nonparametric statistical tests (e.g. Kruskal Wallis, Spearman etc.).

M&M: Please ad SEM images of the emulsion to the manuscript.

Figure 1. Please use lines of different colors in the chart.

P7, L333: Why are references bold?

P8, L347 - 356: Please add discussion concerning Fig.2. Why is carvacrol antioxidant capacity higher? Why is it the lowest for NC? Compare your results to literature data.

Tables 4 and 6: Which statistical test did you use? Tukey or Duncan? Different tests are listed in the M&M than those listed in table explanations.

Comments on the Quality of English Language

The manuscript requires moderate English check and definitely a spell-check.

Author Response

(The authors gave the same response as above.)

Reviewer 4 Report

Comments and Suggestions for Authors

Dear Authors,

all comments are in manuscript.

Best Regards

Comments on the Quality of English Language

Author Response

(The authors gave the same response as above.)

Round 2

Reviewer 1 Report

Comments and Suggestions for Authors

The author has revised the article in detail and given a reasonable explanation. I think it has reached the level of publication.